# Secure Three-Factor Authentication Protocol for Multi-Gateway IoT Environments

**DOI:** 10.3390/s19102358

**Published:** 2019-05-22

**Authors:** JoonYoung Lee, SungJin Yu, KiSung Park, YoHan Park, YoungHo Park

**Affiliations:** 1School of Electronics Engineering, Kyungpook National University, Daegu 41566, Korea; harry250@naver.com (J.L.); darkskiln@naver.com (S.Y.); kisung2@ee.knu.ac.kr (K.P.); 2IT Conversions, Korea Nazarene University, Cheonan, Chungcheongnam-do 31172, Korea

**Keywords:** internet of things, multi-gateway, mutual authentication, cryptanalysis, BAN logic, AVISPA

## Abstract

Internet of Things (IoT) environments such as smart homes, smart factories, and smart buildings have become a part of our lives. The services of IoT environments are provided through wireless networks to legal users. However, the wireless network is an open channel, which is insecure to attacks from adversaries such as replay attacks, impersonation attacks, and invasions of privacy. To provide secure IoT services to users, mutual authentication protocols have attracted much attention as consequential security issues, and numerous protocols have been studied. In 2017, Bae et al. presented a smartcard-based two-factor authentication protocol for multi-gateway IoT environments. However, we point out that Bae et al.’s protocol is vulnerable to user impersonation attacks, gateway spoofing attacks, and session key disclosure, and cannot provide a mutual authentication. In addition, we propose a three-factor mutual authentication protocol for multi-gateway IoT environments to resolve these security weaknesses. Then, we use Burrows–Abadi–Needham (BAN) logic to prove that the proposed protocol achieves secure mutual authentication, and we use the Automated Validation of Internet Security Protocols and Applications (AVISPA) tool to analyze a formal security verification. In conclusion, our proposed protocol is secure and applicable in multi-gateway IoT environments.

## 1. Introduction

Internet of Things (IoT) provides numerous types of services through the internet to exchange data among sensors, embedded systems, and mobile devices. In recent years, IoT environments such as smart buildings, smart factories, smart homes, and smart offices are rapidly becoming a part of our life. A typical IoT architecture consists of heterogeneous micro devices and collects various types of information in real time. However, this is not efficient for practical IoT systems because the communication and computation cost can be increased when the size of IoT networks and the distance between participants are expanded [1,2]. The gateway nodes are deployed to enhance the performance, which provides the ability to communicate with each other efficiently. In a multi-gateway IoT environment, many gateway nodes are deployed and it can process the capability of large-scale IoT networks. IoT environments are also vulnerable to various attacks due to the nature of the open communication channel. Malicious attackers may attempt to insert, delete, and modify the data to obtain users’ sensitive information and masquerade as valid users. Much research has been done to resolve security problems in IoT environments. Secure mutual authentication is a primitive and essential method to provide secure communication and numerous secure mutual authentication protocols for IoT have been presented to provide various security features [2,3,4,5,6,7,8,9,10,11,12,13,14,15,16].

In 2017, Bae et al. [15] proposed a smartcard-based secure authentication protocol in multi-gateway IoT environments to reduce the computational and communication cost. However, we demonstrate that Bae et al.’s protocol is vulnerable to user impersonation, gateway spoofing, and trace and session key disclosure attacks, and does not provide anonymity and a secure mutual authentication. Then, we propose a three-factor authentication protocol that is based on the biometric information of the user, for IoT environments. To analyze the security aspects, we perform an informal security analysis and use Burrows–Abadi–Needham (BAN) logic. Furthermore, we perform a formal security verification using Automated Validation of Internet Security Protocols and Applications (AVISPA) software to check that our protocol can resist man-in-the-middle attacks and replay attacks. We compare the computation cost and security features of our proposed protocol with those of related existing protocols.

The remainder of this paper is as follows. In Section 2 and Section 3, we introduce related works and our preliminary details. In Section 4 and Section 5, we review Bae et al.’s protocol and cryptanalyze its security flaws. Then, we propose a secure three-factor mutual authentication protocol for multi-gateway IoT environments in Section 6. In Section 7, we prove that our proposed protocol provides a secure mutual authentication using BAN logic. We also perform the AVISPA simulation as a formal security verification and compare the computation cost and security properties with related protocols in Section 8 and Section 9. Finally, we conclude with the results of this paper in Section 10.

## 2. Related Works

Various authentication protocols in single server environments have been proposed [3,4,5]. In 2010, Wu et al. [3] presented a novel authentication protocol for the telecare medical information system (TMIS). Their protocol provides a guarantee to legitimate users. However, Debiao et al. [6] demonstrated that Wu et al.’s protocol cannot withstand several attacks such as impersonation, replay, or man-in-the-middle attacks. Debiao et al. proposed a more safe and efficient remote authentication protocol for TMIS. In 2013, Chang et al. proposed a secure authentication protocol that provided users privacy. But, in 2103, Das et al. [7] showed that their protocol cannot provide several security features and proper authentication. Furthermore, these authentication protocols are not suitable for distributed systems that consist of multiple servers, such as IoT environments, because the users who want to access the IoT services have to know as many identities and passwords as the number of servers [8,9]. In addition, the physical performance of a single server has limitations [17], and IoT environments are resource-constrained. Therefore, multi-gateway (multi-server) IoT environments are more efficient and useful than the traditional IoT structure [1,2,10,13,14,15,16].

In 2014, Turkanovic et al. [5] presented an authentication protocol for IoT environments. However, in 2016, Amin and Biswas [10] pointed out that Turkanovic et al.’s protocol does not withstand several attacks such as offline identity and password guessing, impersonation, and stolen smartcard attacks. They also demonstrated that Turkanovic et al.’s protocol has an inefficient authentication phase. Then, Amin and Biswas proposed an authentication protocol for multi-gateway wireless sensor networks. In 2017, Wu et al. [1] proved that Amin and Biswas’s protocol does not resist sensor capture, offline guessing, session key disclosure, impersonation, and desynchronization attacks. They also proved that Amind and Biswas’s protocol does not withstand user tracking attacks and does not achieve mutual authentication. Then, Wu et al. proposed a mutual authentication and key agreement protocol for multi-gateway wireless sensor network in IoT. In the same year, Srinivas et al. [13] also proved that Amin and Biswas’s protocol has security flaws. Srinivas et al. pointed out that sensor devices have low power, limited memory, and limited battery. Thereafter, Srinivas et al. proposed a more secure and efficient remote user authentication protocol for multi-gateway wireless sensor networks that are suitable for IoT environments.

In 2016, Das et al. [10] presented a three-factor multi-gateway-based user authentication protocol for wireless sensor networks. Das et al. suggested the multi-gateway environment for wireless sensor networks because the generalized wireless sensor networks can bring a lot of overhead to the gateway and have more power consumption than multi-gateway-based wireless sensor networks. They demonstrated that their protocol can withstand attacks such as sensor capture, privileged-insider, offline password guessing, and impersonation attacks. However, Wu et al. [1] pointed out that Das et al.’s protocol does not resist user tracking attacks and does not have a same session key for all three participants.

In 2018, Wu et al. [14] proposed an authentication protocol for healthcare systems in multi-gateway wireless medical sensor networks. Their protocol prevents malicious attacks such as patient tracking, insider, and offline guessing attacks. Wu et al. demonstrated that multi-gateway environments are suitable for collecting patients’ health data through wireless health sensors because the gateway in each area collects the information of patients in the area and then sends it to the doctor. They also demonstrated that their protocol is suitable for transferring data with low time and communication costs.

In 2017, Bae et al. [15] proposed a smartcard-based secure authentication protocol in multi-gateway IoT environments to reduce the computational and communication cost. However, their protocol does not resist impersonation, gateway spoofing, traceability, and session key disclosure attacks and does not guarantee secure mutual authentication and anonymity.

## 3. Preliminaries

In this section, we introduce a threat model for cryptanalyzing Bae et al.’s protocol, the fuzzy extraction that we use for the cryptographic system in our authentication protocol, and the system model of our protocol in multi-gateway IoT environments. Finally, we present the notations used in this paper.

### 3.1. Threat Model

We adopt the Dolev–Yao (DY) threat model [18] to analyze Bae et al.’s protocol and our proposed protocol. This model is popularly applied to estimate security. The general assumptions of the DY threat model are as below:An attacker can eavesdrop, delete, modify, or insert the transmitted messages via an insecure channel.An attacker can steal the smartcard or use a lost smartcard to extract the sensitive information stored in the smartcard [19].An attacker can perform various attacks such as trace, impersonation, smartcard lost, man-in-the-middle, replay attacks, and so on.

### 3.2. Fuzzy Extraction

We briefly show a description of the fuzzy extractor [20] that can extract key information from the given biometric data of users. Biometric information is weak to noises and it is hard to reproduce the actual biometrics from biometric templete in common practice. Moreover, the hash function is sensitized to input, so completely different outputs may come out. Because of these problems, we use the fuzzy extractor method [21,22], which is a type of key generating designed to convert noisy data to public information and a secret random string. The fuzzy extractor restores the original biometric information for noisy biometric data using public help information. The algorithms of the fuzzy extractor are as follows:$Generate\left(BIO_{i}\right)=<R_{i},P_{i}>$. This algorithm is for generating key information. It uses biometric data $BIO_{i}$ as an input and then outputs secret key data $R_{i}$, which is a uniformly random string, and a public reproduction $P_{i}$ as a helper string.$Reproduce\left(BIO_{i}^{{}^{\prime }},P_{i}\right)=R_{i}$. This algorithm reproduces the secret data $R_{i}$. The inputs of this algorithm are a noisy biometric $BIO_{i}^{{}^{\prime }}$ and $P_{i}$. The algorithm reproduces the secret biometric key $R_{i}$. To recover the same $R_{i}$, the metric space distance between $BIO_{i}$ and $BIO_{i}^{{}^{\prime }}$ should be within a given error tolerance.

### 3.3. System Model

We introduce a system model of with our proposed protocol for multi-gateway IoT environments. The model consists of three entities: Users, Gateways, and a Control Server. The multi-gateway IoT system model is illustrated in Figure 1.

Users: A user who wants to use the IoT service receives a smartcard from the control server to access the multi-gateway. After registration, login, and authentication, the user has access to use the IoT service. The users’ smartcard can be lost or stolen by an attacker.Gateways: The gateways consist of IoT environments such as smart homes, smart buildings, smart offices, and gateways. We assume that the gateway and IoT environments are connected in advance by a wireless network through a secure authentication. The performance of the gateways is approximately the performance of the server computer.Control Server: The control server is a trusted authentication server with sufficient computation power to compute complicated hash and exclusive functions or store security parameters. The control server stores the identities of the legitimate gateways in advance, and we assume that an attacker can never attack the control server.

### 3.4. Notations

Table 1 shows the notations used in this paper.

## 4. Review of Bae et al.’s Protocol

In this section, we overview Bae et al.’s authentication protocol in multi-gateway IoT environments, which consists of three phases: user and server registration phase, user login and authentication phase, and password update phase. In Bae et al.’s protocol, they assumed that the authentication server CS is trusted.

### 4.1. Registration Phase

If a new user $U_{i}$ or server $S_{j}$ requests registration to the authentication server CS, CS issues the smartcard to $U_{i}$ and sends the necessary value to $S_{j}$. This phase and verifier table is shown in Figure 2 and Table 2, respectively, and the details are as follows.

**Step** **1:**$S_{j}$ requests registration to the CS. $S_{j}$ sends its identity $SID_{j}$ to CS through a secure channel, then CS computes $Serinfor_{j}$ and sends this to $S_{j}$.**Step** **2:**$U_{i}$ chooses the $ID_{i}$, and $PW_{i}$, computes $EncPass_{i}=h(ID_{i}\|h\left(PW_{i}\right))$ and sends the message $\left(ID_{i},EncPass_{i}\right)$ and $UID_{i}$, which is an anonymity value of $U_{i}$, to CS through a closed channel.**Step** **3:**CS receives the message from $U_{i}$. CS computes the secret information value $Userinfor_{i}=h\left(EncsPass_{i}\|x\right)$, stores $\left\{UID_{i},Userinfor_{i},EncPass_{i},h\left(*\right),h\left(x\right)\right\}$ in the smartcard, and stores $Userinfor_{i}$, $UID_{i}$ and statusbit in the verifier table. Then, CS issues the smartcard to $U_{i}$.

### 4.2. Login and Authentication Phase

User $U_{i}$ must send a login request message to $S_{j}$ to use the service of server $S_{j}$. After receiving a request message, $S_{j}$ sends a login request message to control server CS. This phase is illustrated in Figure 3 and the following details.

**Step** **1:**$U_{i}$ inputs his/her $ID_{i}$ and $PW_{i}$ and inputs the smartcard into a smartcard reader. The smartcard computes $EncPass_{i}^{{}^{\prime }}=h(ID_{i}\|h\left(PW_{i}\right))$. Then, the smartcard checks whether $EncPass_{i}\overset{?}{=}EncPass_{i}^{{}^{\prime }}$. If it is equal, $U_{i}$ generates a random number $N_{i1}$ and computes $A_{i}=Userinfor_{i}⊕h\left(x\right)⊕N_{i1}$, $Veru_{i}=h(h\left(x\right)\|N_{i1})$. Then, $U_{i}$ generates Ts to prevent a replay attack. Finally, $U_{i}$ sends the login request message $\left\{UID_{i},A_{i},Veru_{i},Ts\right\}$ to $S_{j}$ through a secure channel.**Step** **2:**If $S_{j}$ receives the login request message, $S_{j}$ generates a random number $N_{i2}$ and computes $B_{i}=Serinfor_{j}⊕N_{i2}$, $Vers_{i}=h(h(SID_{j}\left\|x)\right\|N_{i2})$. Then, $S_{j}$ sends the login request message $\left\{UID_{i},A_{i},Veru_{i},B_{i},Vers_{i},SID_{j},Ts\right\}$ to CS through an open channel.**Step** **3:**After CS receives the login request message from $S_{j}$, CS computes $Ts^{{}^{\prime }}=Ts+1$ and checks $\Delta Ts\geq Ts^{{}^{\prime }}-Ts$ to see whether the login request message is legitimate. If it is valid, CS computes $Serinfor_{j}^{{}^{\prime }}=h(SID_{j}\left\|x\right)$, $N_{i2}^{{}^{\prime }}=Serinfor_{i}^{{}^{\prime }}⊕B_{i}$, $Vers_{i}^{{}^{\prime }}=h(h(SID_{j}\left\|x)\right\|N_{i2}^{{}^{\prime }})$. Then CS compares $Vers_{i}\overset{?}{=}Vers_{i}^{{}^{\prime }}$ to check that the message from $S_{j}$ is valid. If it is equal, CS retrieves $Userinfor_{i}$ from the verifier table using $UID_{i}$ from the login request message. Then, CS computes $N_{i1}^{{}^{\prime }}=Userinfor_{i}⊕h\left(x\right)⊕A_{i}$, $Veru_{i}^{{}^{\prime }}=h(h\left(x\right)\|N_{i1}^{{}^{\prime }})$. If $Veru_{i}\overset{?}{=}Veru_{i}^{{}^{\prime }}$ is correct, CS selects a random number $N_{i3}$ and generates a session key $SK_{i}=h(h(A_{i}\left\|h\left(x\right)\right)⊕h\left(N_{i1}⊕N_{i2}⊕N_{i3}\right))$. CS generates time stamp Ts and computes $C_{i}=N_{i1}⊕N_{i3}⊕h\left(SID_{j}⊕N_{i2}\right)$, $D_{i}=h(A_{i}\left\|h\left(x\right)\right)⊕h\left(SID_{j}⊕N_{i2}\right)$, $E_{i}=N_{i2}⊕N_{i3}⊕h(A_{i}\left\|h\left(x\right)\right)$. Finally, CS sends an authentication message $\left\{C_{i},D_{i},E_{i},Ts\right\}$ to $S_{j}$.**Step** **4:**After $S_{j}$ receives the message from CS, $S_{j}$ computes ${\left(N_{i1}⊕N_{i3}\right)}^{{}^{\prime }}=C_{i}⊕h\left(SID_{j}⊕N_{i2}\right)$, $h(A_{i}{\left||h\left(x\right)\right)}^{{}^{\prime }}=D_{i}⊕h\left(SID_{j}⊕N_{i2}\right)$. $S_{j}$ generates a session key $SK^{{}^{\prime }}=h(h(A_{i}{\left||h\left(x\right)\right)}^{{}^{\prime }}⊕h\left(N_{i1}⊕N_{i2}⊕N_{i3}\right){)}^{{}^{\prime }}$. Then, $S_{j}$ computes $E_{i}=\left(N_{i2}⊕N_{i3}\right)⊕h(A_{i}\|h\left(x\right))$ and sends an authentication message $\left\{E_{i},Ts\right\}$ to $U_{i}$.**Step** **5:**After receiving the message from $S_{j}$, $U_{i}$ computes $Ts^{{}^{\prime }}=Ts+1$ and checks whether $\Delta Ts\geq Ts^{{}^{\prime }}-Ts$. If it is correct, $U_{i}$ computes ${\left(N_{i2}⊕N_{i3}\right)}^{{}^{\prime }}=E_{i}⊕h(A_{i}\|h\left(x\right))$ and generates a session key $SK^{{}^{\prime \prime }}=h(h(A_{i}\left\|h\left(x\right)\right)⊕h\left(N_{i1}⊕N_{i2}⊕N_{i3}\right){)}^{{}^{\prime }}$. Therefore, $U_{i}$, $S_{j}$, and CS generate the same session key, so they can perform the authentication.

### 4.3. Password Change Phase

If $U_{i}$ wants to change his/her password $PW_{i}$ to a new password $PW_{i}^{new}$, the password change phase is performed. This phase is illustrated in Figure 4 and is described as follows.

**Step** **1:**The $U_{i}$ inserts his/her smartcard into a card reader and inputs $ID_{i}$ and $PW_{i}$. Then, $U_{i}$ sends the $\left\{ID_{i},PW_{i}\right\}$ to the smartcard reader through the closed channel.**Step** **2:**After receiving the values from $U_{i}$, the smartcard computes $EncPass_{i}=h(ID_{i}\|h\left(PW_{i}\right))$, $Userinfor_{i}^{{}^{\prime }}=h(EncPass_{i}\left\|x\right)$. The smartcard verifies whether $Userinfor_{i}^{{}^{\prime }}\overset{?}{=}Userinfor_{i}$. If it is equal, the smartcard requests a new password.**Step** **3:**$U_{i}$ inputs a new password $PW_{i}^{new}$ and generates $EncPass_{i}^{new}=h(ID_{i}\|h\left(PW_{i}^{new}\right))$. Then, $U_{i}$ inputs $EncPass_{i}^{new}$ into the smartcard.**Step** **4:**The smartcard computes $Userinfor_{i}^{new}=h(EncPass_{i}^{new}\left\|x\right)$ by using $EncPass_{i}^{new}$. The smartcard updates $Userinfor_{i}$ to $Userinfor_{i}^{new}$ and replaces $Userinfor_{i}$. Finally, the user $U_{i}$ changes his/her password.

## 5. Cryptanalysis of Bae et al.’s Protocol

We analyze the security flaws of Bae et al.’s protocol in this section. Bae et al. asserted that their proposed protocol can prevent various attacks such as user impersonation, server spoofing, and session key disclosure attacks. However, we demonstrate that their protocol does not prevent the following attacks.

### 5.1. User Impersonation Attack

If an attacker $U_{a}$ attempts to impersonate an authorized user $U_{i}$, $U_{a}$ must successfully compute a login request message $\left\{UID_{i},A_{i},Veru_{i},Ts\right\}$. According to Section 3.1, we can assume that $U_{a}$ extracts the values $\left\{UID_{i},Userinfor_{i},EncPass_{i},h\left(x\right)\right\}$ from the smartcard of $U_{i}$ and obtains the transmitted messages over a public channel. After that, $U_{a}$ can impersonate the user in the following steps.

**Step** **1:**$U_{a}$ obtains $\left\{Userinfor_{i},h\left(x\right)\right\}$, $\left\{A_{i},Ts\right\}$ from the smartcard of $U_{i}$ and the previous session, respectively.**Step** **2:**$U_{a}$ computes $N_{i1}=A_{i}⊕Userinfor_{i}⊕h\left(x\right)$ and obtains a random nonce $N_{i1}$. Then $U_{a}$ computes $Veru_{i}=h(h\left(x\right)\|N_{i1})$.**Step** **3:**$U_{a}$ computes $A_{i}=Userinfor_{a}⊕h\left(x\right)⊕N_{a1}$, $Veru_{a}=h(h\left(x\right)\|N_{a1})$. Finally, $U_{a}$ can generate a login request message $\left\{UID_{i},A_{i},Veru_{a},Ts\right\}$ successfully.

### 5.2. Server Spoofing Attack

To obtain the sensitive information of a user, an attacker attempts to impersonate the server. Bae et al. asserted that their protocol can withstand server spoofing attacks. However, we analyze that their protocol does not resist server spoofing. First, an attacker $U_{a}$ obtains message $\left\{E_{i},Ts\right\}$ and extracts the information h(x) from the smartcard of an authorized user. Then, $U_{a}$ can impersonate the server by generating authentication messages in the following steps.

**Step** **1:**$U_{a}$ obtains transmitted messages $\left\{E_{i},Ts\right\}$ in the previous session and extracts h(x) from the smartcard of an authorized user.**Step** **2:**$U_{a}$ computes $h(A_{i}\|h\left(x\right))$ and obtains $\left(N_{i2}⊕N_{i3}\right)$. After that, $U_{a}$ computes $E_{i}=\left(N_{i2}⊕N_{i3}\right)⊕h(A_{i}\|h\left(x\right))$.**Step** **3:**Finally, $U_{a}$ generates authentication messages $\left\{E_{i},Ts\right\}$ successfully.

### 5.3. Session Key Disclosure Attack

Bae et al. demonstrated that their protocol can resist session key disclosure attacks because an attacker cannot compute the values $N_{i1}$, $N_{i2}$, and $N_{i3}$. Furthermore, Bae et al. claimed that the attacker cannot obtain h(x) because the trusted party CS generated h(X). However, we demonstrate that the attacker can compute $N_{i1}$ and $N_{i2}⊕N_{i3}$ and extract h(x) in Section 5.1 and Section 5.2. Thus, the attacker can compute $SK_{i}=h(h(A_{i}\left||h\left(x\right)\right)⊕h\left(N_{i1}⊕N_{i2}⊕N_{i3}\right))$. Therefore, Bae et al.’s protocol is vulnerable to session key disclosure attacks.

### 5.4. Mutual Authentication

In Bae et al.’s protocol, CS computes $Vers_{i}^{{}^{\prime }}$ and $Veru_{i}^{{}^{\prime }}$ to authenticate legitimate $U_{i}$ and $S_{j}$. However, CS cannot generate authentication messages for $U_{i}$ and $S_{j}$. Thus, $U_{i}$ and $S_{j}$ receive the message from CS, but they cannot trust the messages because they cannot check whether the attacker sends the message. Therefore, Bae et al.’s protocol does not achieve mutual authentication.

## 6. A Secure Three-Factor Mutual Authentication Protocol

In this section, we propose a three-factor mutual authentication protocol for multi-gateway IoT environments according to Section 3.3. The proposed protocol consists of three phases: users and gateways registration, login and authentication, and password update.

### 6.1. Registration Phase

First, a gateway $G_{j}$ must register with control server CS to provide their services to users. Then, a new user $U_{i}$ first accesses the control server, and he/she must register with CS. The detailed steps are illustrated in Figure 5 and described as follows.

**Step** **1:**$G_{j}$ requests registration to the CS. $G_{j}$ selects $GID_{j}$ and sends the value to CS through a secure channel, then CS computes $PID_{j}=h(GID_{j}\left\|h(x\|y)\right)$ and sends $PID_{j}$ to $G_{j}$ via a secure channel. $G_{j}$ stores $PID_{j}$ in itself.**Step** **2:**$U_{i}$ chooses the his/her identification $ID_{i}$ and password $PW_{i}$ and imprints biometrics $BIO_{i}$. Then $U_{i}$ generates a random number $a_{i}$, computes $<R_{i},P_{i}>=Gen\left(BIO_{i}\right)$, $HIDi=h(ID_{i}\|a_{i}))$, which is an anonymity value of $U_{i}$, and $HPW_{i}=h(ID_{i}\|PW_{i}\|a_{i})$, and sends the message $\left\{HID_{i},HPW_{i},a_{i}\right\}$ to CS through closed channel.**Step** **3:**After CS receives the message from $U_{i}$, CS computes the secret information value $UI_{i}=h(HID_{i}\|a_{i}\left\|x\right)$, $A_{i}=UI_{i}⊕h\left(HPW_{i}\right)$, $B_{i}=h(UI_{i}\|A_{i})$, and $X_{i}=h(UI_{i}\left\|x\right)$. Then, CS stores $\left\{A_{i},B_{i},X_{i},h\left(*\right)\right\}$ in the smartcard, and stores $UI_{i}$ with $HID_{i}$ in the database. Then CS issues the smartcard to $U_{i}$.**Step** **4:**After receiving the smartcard from CS, $U_{i}$ computes $L_{i}=h(R_{i}\|PW_{i})⊕a_{i}$. Then $U_{i}$ inputs $L_{i}$ and $P_{i}$ in the smartcard.

### 6.2. Login and Authentication Phase

If a user $U_{i}$ wants to use the service of gateway $G_{j}$, $U_{i}$ must send a login request message to $G_{j}$. Then, $G_{j}$ sends a login request message to control server CS. The detailed steps are illustrated in Figure 6 and described as follows.

**Step** **1:**$U_{i}$ inserts the smartcard, his/her $ID_{i}$ and $PW_{i}$, and biometric $BIO_{i}$. The smartcard computes $R_{i}=Rep\left(BIO_{i},Pi\right)$, $a_{i}=L_{i}⊕h(R_{i}\|PW_{i}$), $HID_{i}=h(ID_{i}\|a_{i})$, $HPW_{i}=h(ID_{i}\|PW_{i}\|a_{i})$, $UI_{i}=A_{i}⊕h\left(HPW_{i}\right)$, $B_{i}^{*}=h(UI_{i}\|A_{i})$. Then, the smartcard checks whether $B_{i}^{*}\overset{?}{=}B_{i}$ to check whether the user is legitimate. If it is valid, $U_{i}$ generates a random number $N_{i}$ and computes $C_{i}=UI_{i}⊕N_{i}$, $VU_{i}=h(X_{i}\|N_{i}\|GID_{j})$. Finally, $U_{i}$ sends the login request message $\left\{HID_{i},C_{i},VU_{i}\right\}$ to $G_{j}$ through a public channel.**Step** **2:**After receiving a login request message, $G_{j}$ generates a random number $N_{j}$ and computes $D_{i}=GI_{j}⊕N_{j}$, $VS_{j}=h(GID_{j}\|GI_{j}\|N_{j})$. Then, $G_{j}$ sends the login request message $\left\{HID_{i},C_{i},PID_{j},D_{i},VS_{j}\right\}$ to CS via an open channel.**Step** **3:**After CS receives the login request message from $G_{j}$, CS computes $GI_{j}=h(PID_{j}\left\|h(x\|y)\right)$, $N_{j}=D_{i}⊕GI_{j}$ and compares $VS_{j}^{*}\overset{?}{=}VS_{j}$ to see whether $G_{j}$’s login request message is legitimate. If it is equal, CS retrieves $UI_{i}$ from the verifier table using $HID_{i}$ of the login request message. Then, CS computes $X_{i}=h(UI_{i}\left\|x\right)$, $N_{i}=C_{i}⊕UI_{i},VU_{i}^{*}=h(X_{i}\|N_{i}\|GID_{j})$. Then CS compares $VU_{i}^{*}\overset{?}{=}VU_{i}$ to check that the message from $U_{i}$ is valid. If it is valid, CS generates a random number $N_{c}$ and computes $E_{i}=GI_{j}⊕N_{c}$, $F_{i}=GI_{j}⊕N_{i}$. CS computes $M_{cg}=h(E_{i}\|GI_{j}\|N_{c})$ to mutually authenticate with $G_{j}$ and $M_{cu}=h(X_{i}\|UI_{i}\|N_{i})$ to authenticate with $U_{i}$ and generates a session key $SK=h(N_{i}⊕h(N_{j}\|N_{c}))$. CS updates $HID_{i}$ to $HID_{i}^{new}=h(HID_{i}\|N_{i}\|h(N_{j}\|N_{c}))$ and $UI_{i}$ to $UI_{i}^{new}=h(HID_{i}^{new}\|N_{i}\|UI_{i})$, then replaces $HID_{i}$ and $UI_{i}$. Finally, CS sends the authentication message $\{M_{cg}$, $M_{cu}$, $E_{i}$, $F_{i}\}$ to $G_{j}$.**Step** **4:**After $G_{j}$ receives the authentication message from CS, $G_{j}$ computes $N_{c}=E_{i}⊕GI_{j}$, $M_{cg}^{*}=h(E_{i}\|GI_{j}\|N_{c})$.Then, $G_{j}$ compares $M_{cg}^{*}\overset{?}{=}M_{cg}$ to verify whether the message from CS is legitimate. If it is valid, $G_{j}$ computes $N_{i}=F_{i}⊕GI_{j}$ and generates a session key $SK=h(N_{i}⊕h(N_{j}\|N_{c}))$. Then, $G_{j}$ computes $G_{i}=h(GID_{j}\|N_{i})$, $H_{i}=G_{i}⊕h(N_{j}\|N_{c})$ and sends the authentication message $\left\{H_{i},M_{cu}\right\}$ to $U_{i}$.**Step** **5:**After receiving the message from $G_{j}$, $U_{i}$ computes $M_{cu}^{*}=h(X_{i}\|UI_{i}\|N_{i})$ and verifies whether $M_{cu}^{*}\overset{?}{=}M_{cu}$. If it is valid, $U_{i}$ computes $G_{i}^{*}=h(GID_{j}\|N_{i})$, $h(N_{j}\|N_{c})=H_{i}⊕G_{i}^{*}$ and generates a session key $SK=h(N_{i}⊕h(N_{j}\|N_{c}))$. Therefore, $U_{i}$, $S_{j}$, and CS generate the same session key, so they can perform the authentication. $U_{i}$ updates $HID_{i}$ to $HID_{i}^{new}=h(HID_{i}\|N_{i}\|h(N_{j}\|N_{c}))$ and $UI_{i}$ to $UI_{i}^{new}=h(HID_{i}^{new}\|N_{i}\|UI_{i})$, then replaces $HID_{i}$ and $UI_{i}$. The smartcard updates $A_{i}^{new}=UI_{i}^{new}⊕h\left(HPW\right),B_{i}^{new}=h(UI_{i}^{new}\|A_{i}^{new})$, and $X_{i}^{new}=h(UI_{i}^{new}\|UI_{i})$.

### 6.3. Password Change Phase

If $U_{i}$ wants to change his/her password, $U_{i}$ performs the password change phase without the help of $G_{j}$. The detailed steps of the password change phase are shown in Figure 7 and described as follows.

**Step** **1:**A legitimate user $U_{i}$ inserts the smartcard, his/her $ID_{i}$ and $PW_{i}$, and biometric $BIO_{i}$.**Step** **2:**The smartcard computes $<R_{i},P_{i}>=Gen\left(BIO_{i}\right)$, $a_{i}=L_{i}⊕h(R_{i}\|PW_{i}$), $HPW_{i}=h(ID_{i}\|PW_{i}\|a_{i})$, and $B_{i}^{*}=h(UI_{i}\|A_{i})$. After that, the smartcard compares the $B_{i}^{*}$ with $B_{i}$ stored value. If it is equal, the smartcard requests a new password to $U_{i}$.**Step** **3:**When $U_{i}$ receives the request message from smartcard, $U_{i}$ inputs a new password $PW_{i}^{new}$.**Step** **4:**After receiving the new password from $U_{i}$, the smartcard computes $L_{i}^{new}=a_{i}⊕h(R_{i}\|PW_{i}^{new})$, $HPW_{i}^{new}=h(ID_{i}\|PW_{i}^{new}\|a_{i})$, $A_{i}^{new}=UI_{i}⊕h\left(HPW_{i}^{new}\right)$, and $B_{i}^{new}=h(UI_{i}\|A_{i}^{new})$. Consequently, the smartcard updates the old information $\left\{A_{i},B_{i},L_{i}\right\}$ to new information $\left\{A_{i}^{new},B_{i}^{new},L_{i}^{new}\right\}$.

## 7. Security Analysis

We show that our proposed protocol can prevent various attacks by performing an informal analysis, as mentioned in Section 3.1. We analyze our protocol using Burrows–Abadi–Needham (BAN) logic to prove that our protocol can achieve secure mutual authentication.

### 7.1. Informal Security

To prove that our proposed protocol can prevent various attacks such as trace, smartcard lost, impersonation, off-line guessing, and session key disclosure attacks, we perform an informal security analysis. Additionally, we show that proposed protocol provides anonymity and a secure mutual authentication.

#### 7.1.1. User Impersonation Attack

If a malicious attacker $U_{a}$ attempts to masquerade as a user $U_{i}$, $U_{a}$ can generate a login request message $\left\{HID_{i},C_{i},VU_{i}\right\}$ and message $\left\{H_{i},M_{cu}\right\}$. However, $U_{a}$ cannot compute $HID_{i}$ because $U_{a}$ cannot extract a random number $a_{i}$ from $HID_{i}$. $U_{a}$ cannot retrieve a random number $N_{i}$ because the attacker cannot know secret parameter $UI_{i}$. Thus, $U_{a}$ cannot compute $C_{i},VU_{i}$ because $U_{a}$ cannot extract a random number $N_{i}$. Therefore, our protocol resists user impersonation attack.

#### 7.1.2. Server Spoofing Attack

To impersonate the server, an attacker $U_{a}$ can generate an authentication message $\left\{H_{i},M_{cu}\right\}$. However, $U_{a}$ cannot compute these because $U_{a}$ cannot know the random nonces $N_{i},N_{j},N_{c}$. Furthermore, if $U_{a}$ attempts to impersonate the gateway by using public parameter $GID_{j}$, the control server compares it with the stored identities of the legitimate gateways in advance. Thus, our proposed protocol is secure against server spoofing attacks because $U_{a}$ cannot generate valid messages.

#### 7.1.3. Smartcard Stolen Attack

We assume that an attacker $U_{a}$ can extract the values of the smartcard $\left\{A_{i},B_{i},X_{i},L_{i},h\left(*\right)\right\}$ according to Section 3.1. However, $U_{a}$ cannot obtain sensitive or useful information without the identity, password, and biometrics of the legitimate user because the values stored in the smartcard are safeguarded with a one-way hash function or an XOR operation of $ID_{i},PW_{i},HPW_{i}=h(ID_{i}\|PW_{i}\|a_{i})$. Therefore, our protocol can prevent smartcard stolen attacks.

#### 7.1.4. Trace Attack and Anonymity

In our protocol, an attacker $U_{a}$ cannot know the identity of the users and gateways. The user $U_{i}$ does not send a real identity $ID_{i}$ via the public channels. The user generates and sends a pseudonym identity $HID_{i}=h(ID_{i}\|a_{i})$. Because $HID_{i}$ is a transmitted message via a public channel, $U_{a}$ can obtain this value. Therefore, $U_{i}$ updates it as $HID_{i}^{new}=h(HID_{i}\|N_{i}\|h(N_{j}\|N_{c}))$ for every session to prevent the attack of $U_{a}$. The gateway uses $PID_{j}$, which is generated in the registration phase, instead of $GID_{j}$, so our protocol provides anonymity of users and gateways. In addition, the proposed protocol resists trace attacks because all messages are dynamic for every session.

#### 7.1.5. Man-in-the-Middle Attack and Replay Attack

We assume that attacker $U_{a}$ knows the information transmitted via an insecure channel and information from the smartcard of $U_{i}$ to set up a secure channel with $G_{j}$. However, $U_{a}$ cannot generate a valid login request message, as mentioned. Furthermore, $U_{a}$ cannot impersonate user $U_{i}$ by resending the messages because the messages are refreshed with random numbers $N_{i},N_{j}$, and $N_{c}$. Therefore, our proposed protocol prevents man-in-the-middle attacks and replay attacks.

#### 7.1.6. Off-Line Password Guessing Attack

An attacker $U_{a}$ attempts to guess the password $PW_{i}$ of legitimate user $U_{i}$. If $U_{a}$ can guess the password, $U_{a}$ can compute a series of equations and compute several equations and the valid value with the guessed passwords. However, $U_{a}$ must know the unique biometrics of the user to compute equations. Therefore, it is impossible to guess the user’s password in our protocol.

#### 7.1.7. Desynchronization Attack

For a desynchronization attack, an adversary disturbs the communication of the login and authentication request message. However, CS uses $HID_{i}$ to retrieve $UI_{i}$ after checking message from $G_{j}$, and $HID_{i}$ updates $HID_{i}^{new}$ after authentication of the request message. Furthermore, an attacker disturbs the response communication to desynchronize $HID_{i}^{new}$. Even if the user cannot receive the response message, the user can generate and update $HID_{i}^{new}$. Thus, our proposed protocol can resist desynchronization attacks.

#### 7.1.8. Mutual Authentication

When control server CS receives the login request message from gateway $G_{j}$, CS computes $VS_{j}^{*}$ and $VU_{i}^{*}$ to authenticate user $U_{i}$ and $G_{j}$. If $VS_{j}$ and $VS_{j}^{*}$ are equal, CS authenticates $G_{j}$. Furthermore, CS retrieves $U_{i}$ from a database to an available $VS_{j}$. After that, CS compares $VU_{i}$ and $VU_{i}^{*}$. If they are equal, CS authenticates $U_{i}$. Then, CS computes and sends the login response messages $M_{cg}$ and $M_{cu}$ to authenticate. After receiving $M_{cg}$ from CS, $G_{j}$ computes $M_{cg}^{*}$ and compares $M_{cg}^{*}$ and $M_{cg}$. If they are equal, $G_{j}$ authenticates CS. Finally, $U_{i}$ computes $M_{cu}^{*}$ and checks whether $M_{cu}^{*}\overset{?}{=}M_{cu}$. If it is valid, $U_{i}$ authenticates CS. Therefore, $U_{i}$, $G_{j}$, and CS successfully mutually authenticate. An attacker cannot validate the message, as mentioned in Section 7.1.1 and Section 7.1.2. Moreover, the login request and response messages are refreshed for every session according to Section 7.1.4 and Section 7.1.5. Therefore, our proposed protocol provides secure mutual authentication.

### 7.2. Ban Logic

We perform a formal verification to check that our proposed protocol achieves a secure mutual authentication using BAN logic. Table 3 presents the notation of BAN logic. We show the logical rules of BAN logic in Section 7.2.1. In the following sections, we show the goals, idealized forms, and assumptions of our proposed protocol. In Section 7.2.5, we show that our proposed protocol can provide mutual authentication among $U_{i}$, $G_{j}$, and CS. More details of BAN logic can be found in [23,24].

#### 7.2.1. Rules of Ban Logic

We introduce rules of BAN logic as follows:Message meaning rule:
$$\frac{P\;|\equiv P\overset{K}{↔}Q,\;P⊲{\left\{X\right\}}_{K}}{P\;\left|\equiv Q\;\right|∼X}$$Nonce verification rule:
$$\frac{P\;\left|\equiv \#(X),\;P\;\right|\equiv Q\;|∼X}{P\;\left|\equiv Q\;\right|\equiv X}$$Jurisdiction rule:
$$\frac{P\;\left|\equiv Q\;\right|⟹X,\;P\;\left|\equiv Q\;\right|\equiv X}{P\;|\equiv X}$$Freshness rule:
$$\frac{P\;|\equiv \#(X)}{P\;|\equiv \#\left(X,Y\right)}$$Belief rule:
$$\frac{P\;|\equiv \left(X,Y\right)}{P\;|\equiv X}$$

#### 7.2.2. Goals

We present the following goals to prove that our protocol achieves secure mutual authentication:**Goal** **1:**$G_{j}\left|\equiv CS\right|\equiv \left(N_{c},N_{i}\right)$,**Goal** **2:**$G_{j}|\equiv \left(N_{c},N_{i}\right)$,**Goal** **3:**$CS|\equiv G_{j}|\equiv \left(N_{i},N_{j}\right)$,**Goal** **4:**$CS|\equiv (N_{i},N_{j})$,**Goal** **5:**$U_{i}\left|\equiv G_{j}\right|\equiv \left(N_{c},N_{i}\right)$,**Goal** **6:**$U_{i}|\equiv \left(N_{j},N_{c}\right)$

#### 7.2.3. Idealized Forms

$Msg_{1}:$
$U_{i}\to G_{j}:{\left(HID_{i},N_{i},x,GID_{j}\right)}_{UI_{i}}$
$Msg_{2}:$
$G_{j}\to CS:{\left(HID_{i},N_{i},x,GID_{j},N_{j}\right)}_{GI_{j}}$
$Msg_{3}:$
$CS\to G_{j}:{\left(N_{c},N_{i},UI_{i},x\right)}_{GI_{j}}$
$Msg_{4}:$
$G_{j}\to U_{i}:{\left(N_{c},N_{j},UI_{i},GID_{j},x\right)}_{N_{i}}$


#### 7.2.4. Assumptions

To achieve the BAN logic proof, we make the following assumptions about the initial state of our proposed protocol:$A_{1}:$$G_{j}|\equiv \left(U_{i}\overset{UI_{i}}{⟷}G_{j}\right)$$A_{2}:$$G_{j}|\equiv \#\left(N_{i}\right)$$A_{3}:$$CS|\equiv (G_{j}\overset{GI_{j}}{⟷}CS)$$A_{4}:$$CS|\equiv \#(N_{j},N_{i})$$A_{5}:$$G_{j}|\equiv \left(G_{j}\overset{GI_{j}}{⟷}CS\right)$$A_{6}:$$U_{i}|\equiv \left(U_{i}\overset{N_{i}}{⟷}G_{j}\right)$$A_{7}:$$U_{i}|\equiv \#\left(N_{j}\right)$$A_{8}:$$CS|\equiv G_{j}\Rightarrow \left(CS\overset{GI_{j}}{⟷}G_{j}\right)$

#### 7.2.5. Proof Using Ban Logic

The following steps are the main proofs using BAN rules and assumptions:**Step** **1:**According to $Msg_{1}$, we can get
$$S_{1}:G_{j}⊲{\left(HID_{i},N_{i},x,GID_{j}\right)}_{UI_{i}}.$$**Step** **2:**From $A_{1}$ and $S_{1}$, we apply the message meaning rule to obtain
$$S_{2}:G_{j}|\equiv U_{i}\;{\left(HID_{i},N_{i},x,GID_{j}\right)}_{UI_{i}}.$$**Step** **3:**From $A_{2}$ and $S_{2}$, we apply the freshness rule to obtain
$$S_{3}:G_{j}|\equiv \#{\left(HID_{i},N_{i},x,GID_{j}\right)}_{UI_{i}}.$$**Step** **4:**From $S_{2}$ and $S_{3}$, we apply the nonce verification rule to obtain
$$S_{4}:G_{j}|\equiv U_{i}\equiv {\left(HID_{i},N_{i},x,GID_{j}\right)}_{UI_{i}}.$$**Step** **5:**From $S_{4}$, we apply the belief rule to obtain
$$S_{5}:G_{j}\left|\equiv U_{i}\right|\equiv {\left(N_{i}\right)}_{UI_{i}}.$$**Step** **6:**According to $Msg_{2}$, we can get
$$S_{6}:CS⊲{\left(HID_{i},N_{i},x,GID_{j},N_{j}\right)}_{GI_{j}}.$$**Step** **7:**From $A_{3}$ and $S_{6}$, we apply the message meaning rule to obtain
$$S_{7}:CS|\equiv G_{j}\;{\left(HID_{i},N_{i},x,GID_{j},N_{j}\right)}_{GI_{j}}.$$**Step** **8:**From $A_{4}$ and $S_{7}$, we apply the freshness rule to obtain
$$S_{8}:CS|\equiv \#{\left(HID_{i},N_{i},x,GID_{j},N_{j}\right)}_{GI_{j}}.$$**Step** **9:**From $S_{7}$ and $S_{8}$, we apply the nonce verification rule to obtain
$$S_{9}:CS|\equiv G_{j}|\equiv {\left(HID_{i},N_{i},x,GID_{j},N_{j}\right)}_{GI_{j}}.$$**Step** **10:**From $S_{9}$, we apply the belief rule to obtain
$$S_{10}:CS|\equiv G_{j}|\equiv {\left(N_{i},N_{j}\right)}_{GI_{j}}.(\mathrm{Goal}3)$$**Step** **11:**According to $Msg_{2}$, we can get
$$S_{11}:G_{j}⊲{\left(N_{c},N_{i},UI_{i},x\right)}_{GI_{j}}.$$**Step** **12:**From $A_{5}$ and $S_{11}$, we apply the message meaning rule to obtain
$$S_{12}:G_{j}|\equiv CS\;{\left(N_{c},N_{i},UI_{i},x\right)}_{GI_{j}}.$$**Step** **13:**From $A_{6}$ and $S_{12}$, we apply the freshness rule to obtain
$$S_{13}:G_{j}|\equiv \#{\left(N_{c},N_{i},UI_{i},x\right)}_{GI_{j}}.$$**Step** **14:**From $S_{12}$ and $S_{13}$, we apply the nonce verification rule to obtain
$$S_{14}:G_{j}\left|\equiv CS\right|\equiv {\left(N_{c},N_{i},UI_{i},x\right)}_{GI_{j}}.$$**Step** **15:**From $S_{14}$, we apply the belief rule to obtain
$$S_{15}:G_{j}\left|\equiv CS\right|\equiv {\left(N_{c},N_{i}\right)}_{GI_{j}}.(\mathrm{Goal}1)$$**Step** **16:**According to $Msg_{4}$, we can obtain
$$S_{16}:U_{i}⊲{\left(N_{c},N_{j},UI_{i},GID_{j},x\right)}_{N_{i}}.$$**Step** **17:**From $A_{6}$ and $S_{16}$, we apply the message meaning rule to obtain
$$S_{17}:U_{i}|\equiv G_{j}\;{\left(N_{c},N_{j},UI_{i},GID_{j},x\right)}_{N_{i}}.$$**Step** **18:**From $A_{7}$ and $S_{17}$, we apply the freshness rule to obtain
$$S_{18}:U_{i}|\equiv \#G_{j}\;{\left(N_{c},N_{j},UI_{i},GID_{j},x\right)}_{N_{i}}.$$**Step** **19:**From $S_{17}$ and $S_{18}$, we apply the nonce verification rule to obtain
$$S_{19}:U_{i}\left|\equiv G_{j}\right|\equiv {\left(N_{c},N_{j},UI_{i},GID_{j},x\right)}_{N_{i}}.$$**Step** **20:**From $S_{19}$, we apply the belief rule to obtain
$$S_{20}:U_{i}\left|\equiv G_{j}\right|\equiv {\left(N_{c},N_{j}\right)}_{N_{i}}.(\mathrm{Goal}5)$$**Step** **21:**From $S_{10}$ and $A_{8}$, we apply the jurisdiction rule to obtain
$$S_{21}:CS|\equiv (N_{i},N_{j}).(\mathrm{Goal}4)$$**Step** **22:**From $S_{15}$ and $A_{9}$, we apply the jurisdiction rule to obtain
$$S_{22}:G_{j}|\equiv \left(N_{c},N_{i}\right).(\mathrm{Goal}2)$$**Step** **23:**From $S_{20}$ and $A_{10}$, we apply the jurisdiction rule to obtain
$$S_{23}:U_{i}|\equiv \left(N_{c},N_{i}\right).(\mathrm{Goal}6)$$

We show that the proposed protocol can provide secure mutual authentication between $U_{i}$, $G_{j}$, and CS based on goals 1–6.

## 8. Formal Verification Using Avispa

We present a formal verification of our proposed protocol using the AVISPA tool based on the High-Level Protocol Specification Language (HLPSL) code [25]. AVISPA is one of the widely used verification tools to check that protocols are secure against man-in-the-middle attacks and replay attacks. Numerous studies have been simulated using the AVISPA tool [26,27,28]. We will shortly describe AVISPA and show the HLPSL specifications of our proposed protocol. Then, we will assert that the proposed protocol can resist replay and man-in-the-middle attacks through the results of the AVISPA simulation.

### 8.1. Description of Avispa

AVISPA performs security verification through four back-ends consisting of Constraint-Logic-based Attack Searcher (CL-AtSe) [29], On-the-Fly Model-Checker (OFMC) [30], Tree Automate-Based Protocol Analyzer (TA4SP), and SAT-Based Model-Checker (SATMC). HLPSL specification is translated into intermediate format (IF) by an hlpsl2if translator. IF is converted to the output format (OF), which is produced using the four back-ends as mentioned above. But usually, CL-Atse and OFMC are used for verification. AVISPA has several functions that are mentioned below for analyzing protocols. More details on AVISPA can be found in [31,32].

secret(A,id,B): id denotes an information *A* that is only known to *B*.witness(A,B,id,E): id denotes a weakness authentication factor *E* that is used by *A* to authenticate *B*.request(A,B,id,E): id denotes a strong authentication factor. *B* requests *A* for *E* to authenticate.

### 8.2. Hlpsl Specifications of Our Protocol

Our protocol has three basic roles which are denoted by entities that have been specified according to HLPSL: UA denotes a user, GA denotes a gateway, and CS denotes a control server. The role of session and environments are shown in Figure 8. In the session, we describe participants. In environments, intruder knowledge is defined, and four secrecy goals and four authentication goals are described. The HLPSL specifications of role UA are shown in Figure 9, and the details are as follows.

At transition 1, UA starts the registration phase with a start message in state value 0 and then updates the state from 0 to 1. UA sends the registration message $\left\{HID_{i},HPW_{i},a\right\}$ to CS through a closed channel. At transition 2, UA receives the smartcard from CS, then it updates the state from 1 to 2. In state value 2, UA generates the random number $N_{i}$, sends the login request message $\left\{HID_{i},C_{i},VU_{i}\right\}$ to GA via an insecure channel, and declares $witness(UA,CS,us\_cs\_ni,N_{i})$, which means that $N_{i}$ denotes a weakness authentication factor. At transition 3, UA receives the login response message from GA. After that, UA changes the state value from 2 to 3, generates the session key, and declares $request(UA,CS,cs\_ua\_mcu,N_{c})$. The specifications of role GA and CS are similar and shown in Figure 10 and Figure 11.

### 8.3. Results of Avispa Simulation

The results of AVISPA simulation through OFMC and CL-AtSe verification are shown in Figure 12. The OFMC and CL-AtSe back-ends check whether our proposed protocol can resist replay attacks and man-in-the-middle attacks. The OFMC verification shows that search time is 12 s for visiting 1040 nodes, and the CL-AtSe verification analyzes 3 states with 0.13 s to translate. Because the summary part of OFMC and CL-AtSe indicates that the protocol is SAFE, our proposed protocol is secure against replay and man-in-the-middle attacks.

## 9. Performance Analysis

In this section, we show the comparison of computation cost, communication cost, and security features among our proposed protocol and other IoT-related protocols.

### 9.1. Computation Cost

We compare the computational overhead between our proposed protocol and other related protocols. We define some notations for convenience of comparison.

$T_{me}$: The times for modular exponential operation (≈0.522 s [33,34])$T_{h}$: The times for one-way hash operation (≈0.0005 s [33,34])$T_{f}$: The times for fuzzy extraction operation (≈0.063075 s [34,35])

Table 4 shows the results of the comparison. In multi-gateway environments, it is important to reduce the computation cost of gateway nodes because the gateway nodes process a large amount of information. Although the total computation cost of our proposed protocol is higher than other related protocols, it is similar to [15] in terms of gateway nodes. Therefore, our proposed protocol is suitable for practical IoT environments.

### 9.2. Communication Cost

We have compared the communication overheads at the login and authentication phase of our proposed protocol and other related protocols in Table 5. We assume that the acknowledgment message and the one-way hash function, the timestamp, random number, and identity all are 160 bits. Additionally, we assume that the AES (Advanced Encryption Standard) key is 512 bits [33]. According to the results, our proposed protocol has more efficiency than other related protocols.

### 9.3. Security Properties

Table 6 shows the security comparisons among the proposed protocol and other related protocols based on IoT environment. Our proposed protocol can resist more attacks than other related protocols. Furthermore, our proposed protocol provides anonymity and achieves mutual authentication. Therefore, we demonstrate that the proposed protocol is more safe than other related protocols and satisfies the security requirements of IoT environments.

## 10. Conclusions

IoT is becoming a part of our life and helps people to easily communicate data and comfortably obtain mobile services. However, data scalability, unsolved security problems, and malicious attacks can limit the widespread extension of IoT services. The gateway nodes must process a large amount of information to provide IoT services to users. Thus, reducing the computation cost of gateways is a very important issue, and users and gateways should verify each other’s legitimacy with the aid of a control server to provide authorized and secure communication. In this paper, we demonstrated the security weaknesses of Bae et al.’s protocol. We showed that their protocol is vulnerable to user impersonation attacks, gateway spoofing attacks, session key disclosure attacks, offline password guessing attacks, and does not provide secure mutual authentication. Moreover, we proposed a multi-factor mutual authentication protocol for multi-gateway IoT environments with better security functionality than that of Bae et al.’s protocol. We also proved the security of the proposed protocol using BAN logic and the AVISPA tool.

## Figures and Tables

**Figure 1 sensors-19-02358-f001:**
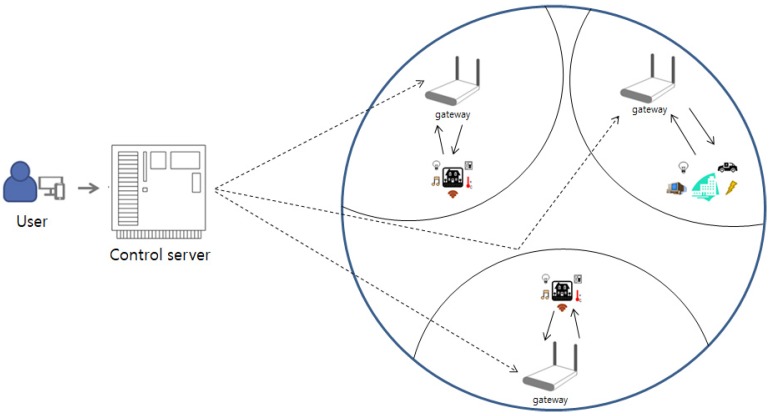
System model of our protocol in multi-gateway IoT environments.

**Figure 2 sensors-19-02358-f002:**
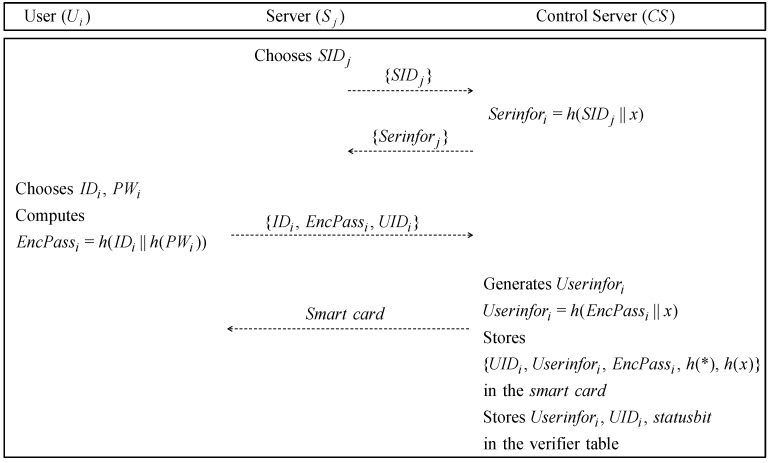
Registration phase of Bae et al.’s protocol.

**Figure 3 sensors-19-02358-f003:**
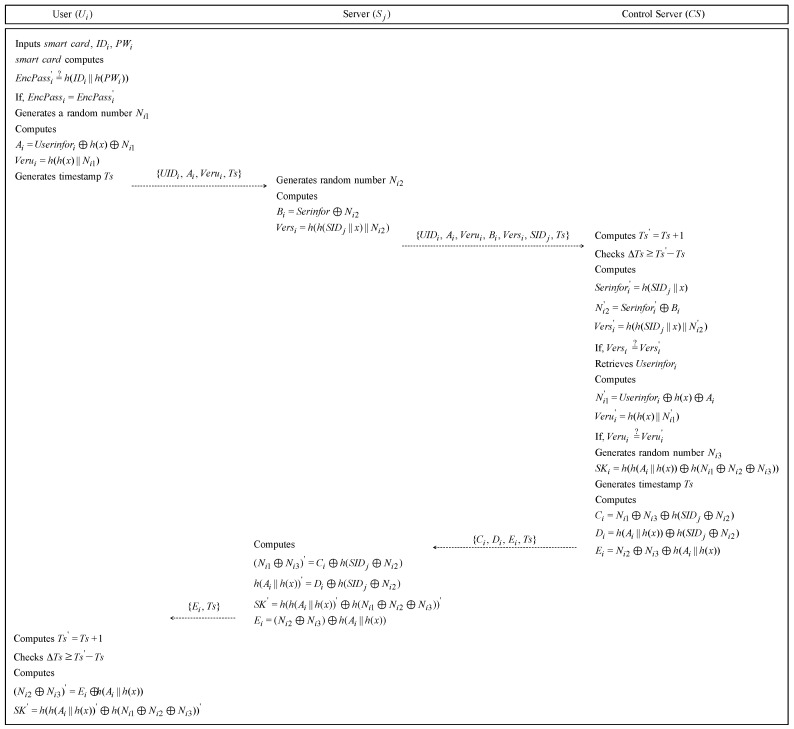
Login and authentication phase of Bae et al.’s protocol.

**Figure 4 sensors-19-02358-f004:**
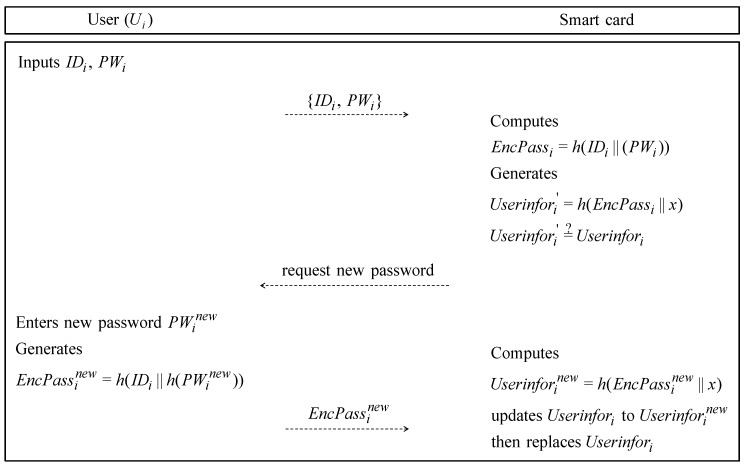
Password change phase of Bae et al.’s protocol.

**Figure 5 sensors-19-02358-f005:**
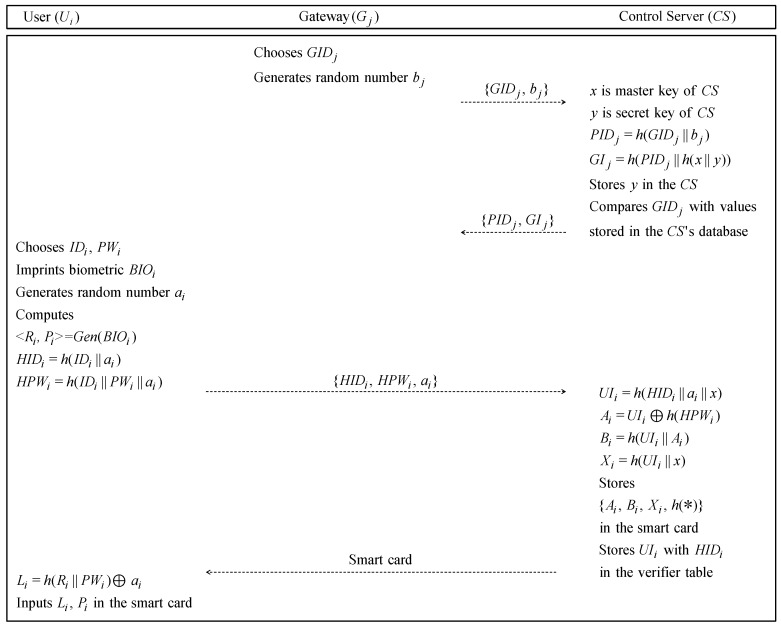
Registration phase of our proposed protocol.

**Figure 6 sensors-19-02358-f006:**
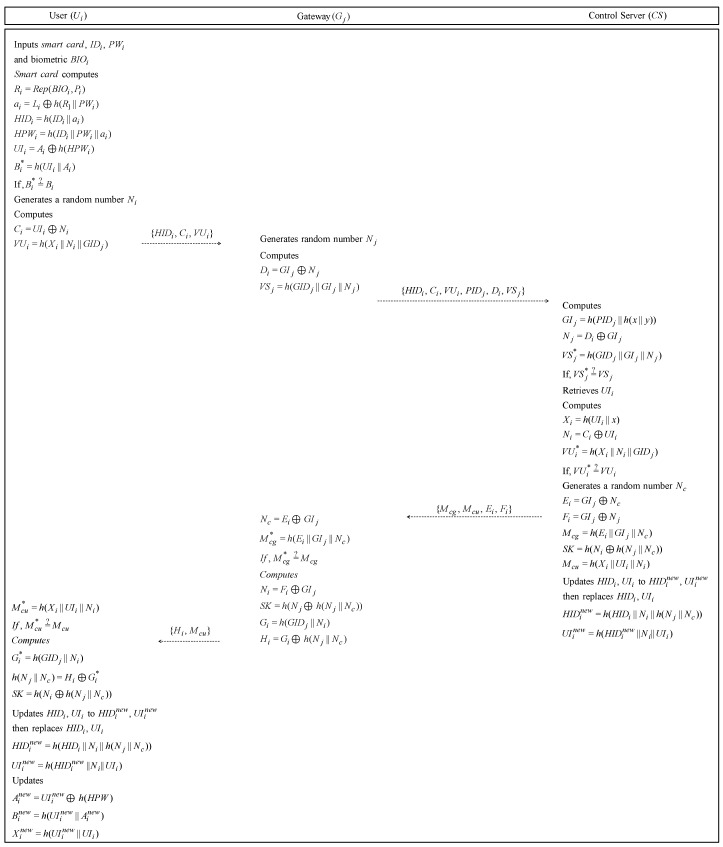
Login and authentication phase of our proposed protocol.

**Figure 7 sensors-19-02358-f007:**
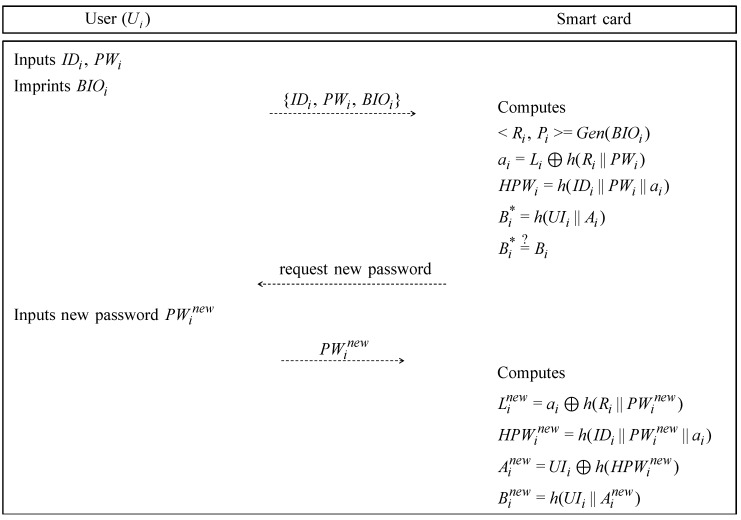
Password change phase of our proposed protocol.

**Figure 8 sensors-19-02358-f008:**
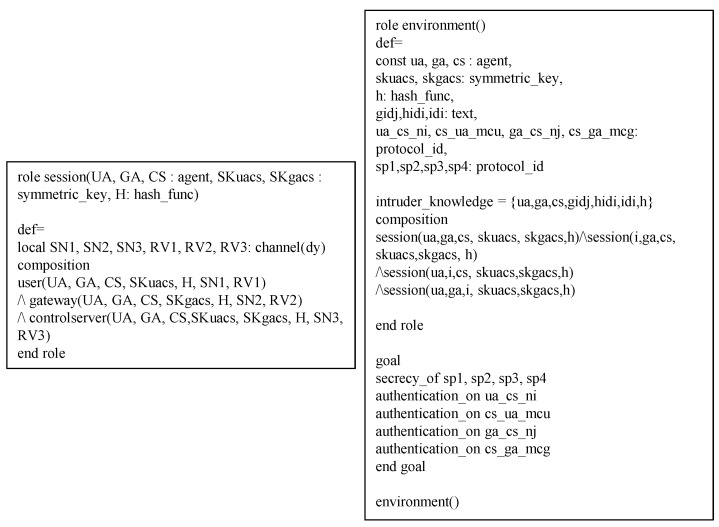
Specification of session and environments.

**Figure 9 sensors-19-02358-f009:**
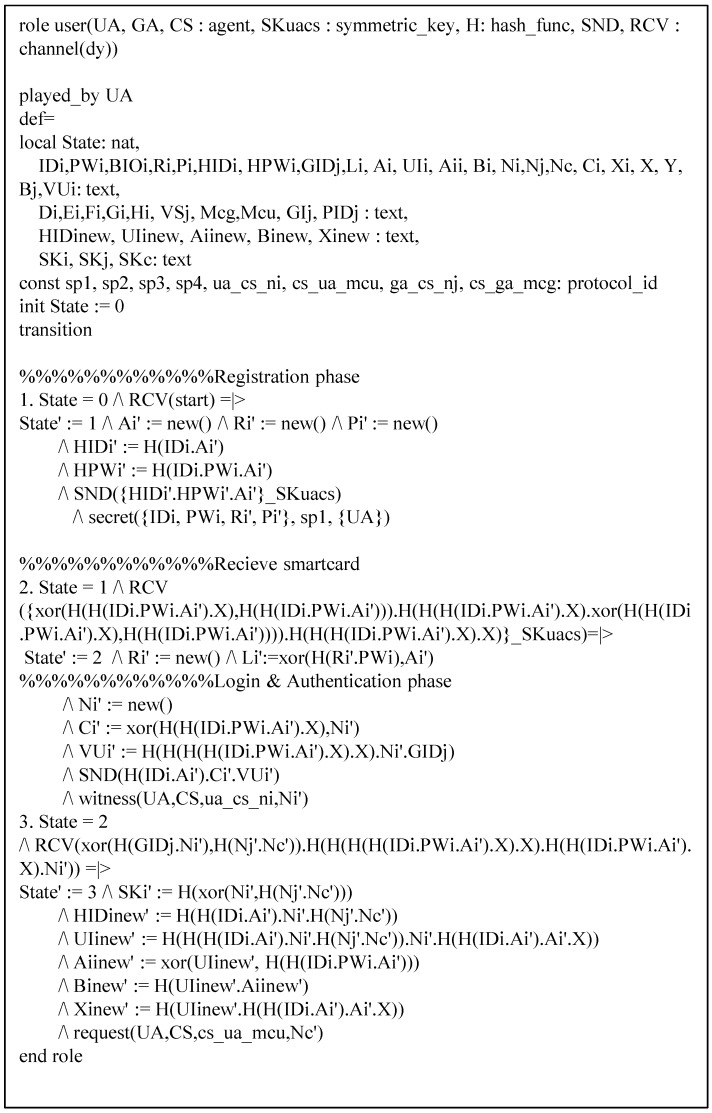
Specification of user.

**Figure 10 sensors-19-02358-f010:**
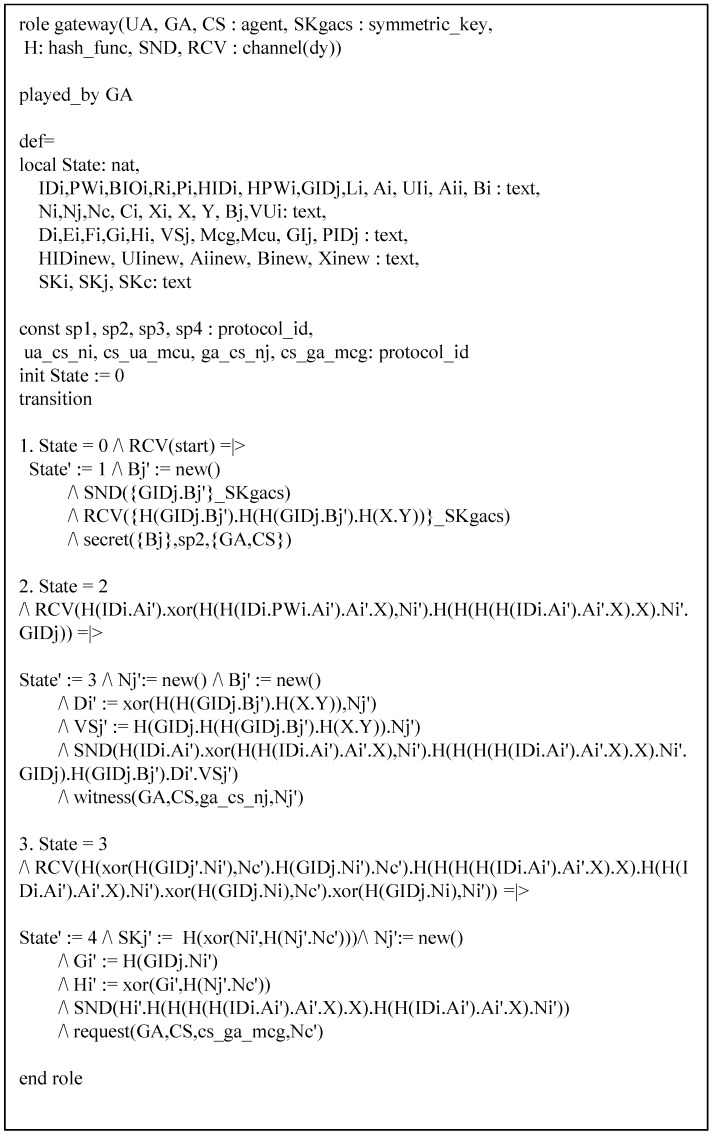
Specification of gateway.

**Figure 11 sensors-19-02358-f011:**
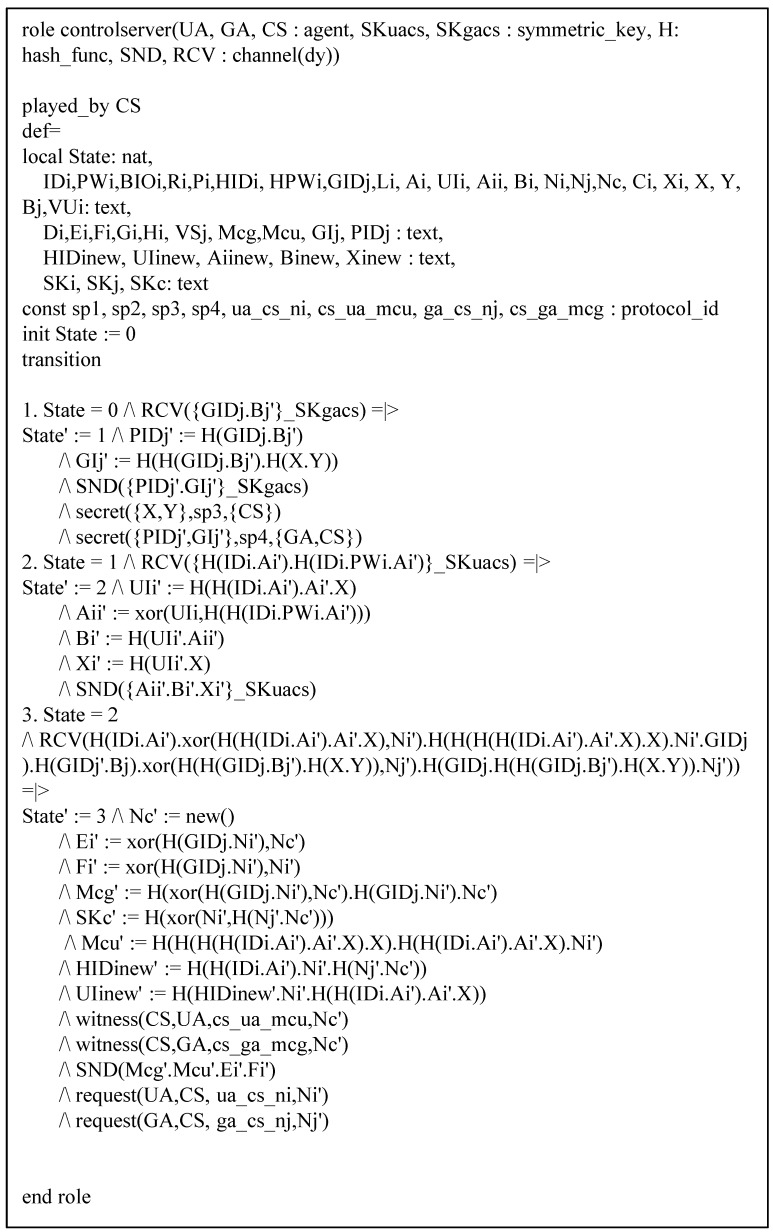
Specification of control server.

**Figure 12 sensors-19-02358-f012:**
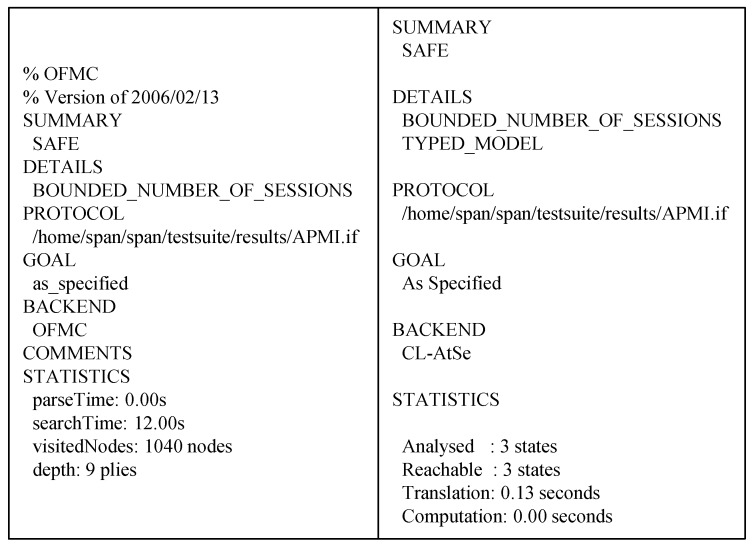
The result of Automated Validation of Internet Security Protocols and Applications (AVISPA) simulation using OFMC and CL-AtSe.

**Table 1 sensors-19-02358-t001:** Notations.

Notations	Meanings
$U_{i}$	i-th user
$S_{j}$	j-th server
CS	Control server
$ID_{i}$	Identity of $U_{i}$
$SID_{j}$	Identity of $S_{j}$
$PW_{i}$	Password of $U_{i}$
*x*	Master secret key chosen by CS
Ts	Timestamp
$N_{i1}$	Random number generated by $U_{i}$’s smartcard
$N_{i2}$	Random number generated by $S_{j}$
$N_{i3}$	Random number generated by CS
SK	Common session key shared among $U_{i}$, $S_{j}$, and CS
h(*)	Collision-resistant one-way hash function

**Table 2 sensors-19-02358-t002:** The verifier table.

User-Verifier	Anonymity Value	Status-Bit
$Userinfor_{1}$	$U_{1}$	0/1
$Userinfor_{2}$	$U_{2}$	0/1
…	…	…
$Userinfor_{i}$	$U_{i}$	0/1

**Table 3 sensors-19-02358-t003:** Notations of Burrows–Abadi–Needham (BAN) logic.

Notations	Meaning
P|≡X	*P* believes the statement *X*
#X	The statement *X* is fresh
P⊲X	*P* sees the statement *X*
P|X	*P* once said *X*
P⇒X	*P* controls the statement *X*
$<X{>}_{Y}$	Formula *X* is combined with formula *Y*
${\left\{X\right\}}_{K}$	Formula *X* is encrypted by the key *K*
$P\overset{K}{↔}Q$	*P* and *Q* communicate using *K* as the shared key
SK	Session key used in the current authentication session

**Table 4 sensors-19-02358-t004:** Computation cost of the login and authentication phase.

Protocols	User	Gateway	Control Server	Total Cost
Turkanovic et al. [5]	7$T_{h}$	5$T_{h}$	7$T_{h}$	19$T_{h}\left(0.0095s\right)$
Wu et al. [3]	2$T_{me}$ + 4$T_{h}$	-	1 $T_{me}$ + 4$T_{h}$	3$T_{me}$+ 8$T_{h}\left(1.57s\right)$
Amin and Biswas Case-1 [10]	7$T_{h}$	5$T_{h}$	8$T_{h}$	20$T_{h}\left(0.01s\right)$
Amin and Biswas Case-2 [10]	8$T_{h}$	5$T_{h}$	7$T_{h}$	20$T_{h}\left(0.01s\right)$
Bae et al. [15]	5$T_{h}$	6$T_{h}$	10$T_{h}$	21$T_{h}\left(0.0105s\right)$
Ours	1$T_{f}$+14$T_{h}$	5$T_{h}$	9$T_{h}$	1$T_{f}$+ 28$T_{h}\left(0.07707s\right)$

XOR operation is negligible compared to other operations.

**Table 5 sensors-19-02358-t005:** Communication cost.

Protocols	Communication Cost
Turkanovic et al. [5]	4000 bits
Wu et al. [3]	2368 bits
Amin and Biswas Case-1 [10]	2080 bits
Amin and Biswas Case-2 [10]	3520 bits
Bae et al. [15]	2720 bits
Ours	2400 bits

**Table 6 sensors-19-02358-t006:** Security properties.

Security Property	Turkanovic et al. [5]	Wu et al. [3]	Amin and Biswas [10]	Bae et al. [15]	Ours
User impersonation attack	x	x	o	x	o
Server spoofing attack	o	x	x	x	o
Smartcard stolen attack	x	x	x	x	o
Trace attack	x	x	x	x	o
Off-line password guessing attack	x	o	x	o	o
Replay attack	o	o	o	o	o
Man-in-the-middle attack	o	o	o	o	o
Desynchronization attack	-	-	x	-	o
Anonymity	x	x	x	o	o
Mutual authentication	x	x	o	x	o

x: does not prevent the property; o: prevents the property; -: does not concern the property.

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
