# Peer review of "Secure Three-Factor Authentication Protocol for Multi-Gateway IoT Environments"

_sensors, 2019, doi:10.3390/s19102358_

Round 1

Reviewer 1 Report

     In this paper, the authors point out that Bae et al.’s protocol is vulnerable to user impersonation attacks, gateway, spoofing attacks, session key disclosure, and cannot provide a mutual authentication. And a three-factor mutual authentication protocol is proposed for multi-gateway IoT environments to resolve these security weaknesses.

      The method consists of registration phase, login and authentication phase and password change phase. Security analysis proves that the protocol can resist user impersonation attack, server spoofing attack, smart card stolen attack, trace attack and Man-in-the-middle attack. Additionally, the authors use Burrows-Abadi-Needham (BAN) logic and the Automated Validation of Internet Security Protocols and Applications (AVISPA) tool to verify the security of the proposed method.

(1) In the introduction of this paper, the background is not sufficient and there is no related work. Please add the section of related work.

(2) This paper points out that multi-gateway (multi-server) IoT environments are more efficient and useful than the traditional IoT structure. Please present the detailed description.

(3) In the password change phase of the proposed protocol, smart card updates the old parameters to the new parameters with new ID and new password. But UIi and HIDi stored in control server are not updated. Moreover, in exiting protocols, control server is not involved in the password change phase. Please improve the proposed protocol and enhance the performance.

(4) In the performance analysis, the authors only present a table of computation cost and the detailed analysis is not shown. Moreover, the data in the table is not derived from the experiment. So,the result is not convinced enough. The authors should present the experimental tool and add corresponding figures appropriately.

Author Response

  We thank for your interest in our work and for constructive comments that will greatly improve the manuscript and we have tried to do our best to respond to the points raised. We have checked the all your comments and have made necessary changes accordingly to your indications. The specific responses are attached

Reviewer 2 Report

The main assumption in this paper is that the protocol proposed by Bae et al is insecure, since in their attacker model the attacker can extract information saved from a smartcard, which has been well adopted by other research papers. Compared to existing solutions, the authors, besides knowing password (something user knows), requires users to use biometry for regular protocol execution. Compared to other solutions, introducing biometry reduces the risk of smartcard stolen attack. Therefore, the authors should provide with example of using the proposed solution, especially in the scenario where biometry is combined with smartcards.

The paper is well written and organized. 

The authors should double check english to correct minor spelling errors: eg. can impersonates the user in the following steps -> can impersonate the user in the following steps

In the description of Bae et al protocol the authors introduce to P_i as Password of U_i. However, in their proposal of the protocol P_i denotes public help information generated as a result of fuzzy extractor, while password is denoted with PW_i. Authors are advised to use different notation.

Also, should P_i - public help information be saved somewhere on the (smartcard) device, so user could extract secret R_i once BIO information is inserted R_i = Rep(BIO||P_i).

Author Response

  We thank for your interest in our work and for constructive comments that will greatly improve the manuscript and we have tried to do our best to respond to the points raised. We have checked the all your comments and have made necessary changes accordingly to your indications.
